# Mitochondrial Phylogenomics of Tenthredinidae (Hymenoptera: Tenthredinoidea) Supports the Monophyly of Megabelesesinae as a Subfamily

**DOI:** 10.3390/insects12060495

**Published:** 2021-05-26

**Authors:** Gengyun Niu, Sijia Jiang, Özgül Doğan, Ertan Mahir Korkmaz, Mahir Budak, Duo Wu, Meicai Wei

**Affiliations:** 1College of Life Sciences, Jiangxi Normal University, Nanchang 330022, China; niug@jxnu.edu.cn (G.N.); duowu27@126.com (D.W.); 2College of Forestry, Beijing Forestry University, Beijing 100083, China; jsj_1998@163.com; 3Department of Molecular Biology and Genetics, Faculty of Science, Sivas Cumhuriyet University, Sivas 58140, Turkey; odogan@cumhuriyet.edu.tr (Ö.D.); budakmah@gmail.com (M.B.); ekorkmaz@cumhuriyet.edu.tr (E.M.K.)

**Keywords:** mitochondrial DNA, Asia, coevolution, phylogenetic systematics, sawfly, feeding adaptation

## Abstract

**Simple Summary:**

Tenthredinidae is the most speciose family of the paraphyletic ancestral grade Symphyta, including mainly phytophagous lineages. The subfamilial classification of this family has long been problematic with respect to their monophyly and/or phylogenetic placements. This article reports four complete sawfly mitogenomes of *Cladiucha punctata*, *C. magnoliae*, *Megabeleses magnoliae*, and *M. liriodendrovorax* for the first time. To investigate the mitogenome characteristics of Tenthredinidae, we also compare them with the previously reported tenthredinid mitogenomes. To explore the phylogenetic placements of these four species within this ecologically and economically important sawfly family, we perform the mitophylogenomics reconstruction and divergence time estimation analyses using a mitogenome dataset of 45 species of the superfamily Tenthredinoidea. Although these newly reported four species have currently classified in the subfamily of Allantinae, the obtained tree topology reveals the sister-group placement of *Cladiucha* and *Megabeleses* outside of Blennocampinae, Heterarthrinae, Tenthredininae, and the rest of Allantinae as a distinct lineage. In conjunction with the occurrence of morphological and molecular synapomorphic characters, apparent matches the reported divergence times of their host plants from Magnoliaceae—supporting the proposal of Megabelesesinae as a subfamily. This study represents a broad framework and valuable information for future research on this small subfamily.

**Abstract:**

Tenthredinidae represents one of the external feeders of the most diverse superfamily, Tenthredinoidea, with diverse host plant utilization. In this study, four complete mitochondrial genomes (mitogenomes), those of *Cladiucha punctata*, *Cladiucha magnoliae*, *Megabeleses magnoliae*, and *Megabeleses liriodendrovorax*, are newly sequenced and comparatively analyzed with previously reported tenthredinid mitogenomes. The close investigation of mitogenomes and the phylogeny of Tenthredinidae leads us to the following conclusions: The subfamilial relationships and phylogenetic placements within Tenthredinidae are mostly found to be similar to the previously suggested phylogenies. However, the present phylogeny supports the monophyly of Megabelesesinae as a subfamily, with the sister-group placement of *Cladiucha* and *Megabeleses* outside of Allantinae. The occurrence of the same type of tRNA rearrangements (MQI and ANS1ERF) in the mitogenomes of Megabelesesinae species and the presence of apomorphic morphological characters also provide robust evidence for this new subfamily. The divergence and diversification times of the subfamilies appear to be directly related to colonization of the flowering plants following the Early Cretaceous. The origin time and diversification patterns of Megabelesesinae were also well matched with the divergence times of their host plants from Magnoliaceae.

## 1. Introduction

The exclusively herbivorous larvae of Tenthredinidae are mostly simple external feeders that feed on a wide variety of leaves, but a few are leaf miners, gall causers, or stem borers [1]. Some species of tenthredinids are economically essential pests on both deciduous and resinous trees as forest defoliators; they feed on apple, plum, pear, red currant, and gooseberry as orchard pests; and on rose and columbine as ornamental-plant pests [2,3]. The family is almost globally distributed and is more diverse in the Northern Hemisphere—in fact, its members are absent or rare in Antarctica, New Zealand, and Australia [2]. The observed distribution pattern is affected by adults who are slow fliers; this trait limits the distribution of the larvae of this diverse family on host plants. In addition to contributing to the overall distribution pattern, the host plant preferences of tenthredinid larvae have also led to an increase in speciation events by host switching, particularly in sublineages that feed on only some angiosperms [4].

This family is the largest and a considerably complex family of Symphyta, comprising 76.8% of the described species of the most diverse superfamily of ancestrally herbivorous hymenopterans, Tenthredinoidea, and approximately 65% of known Symphytan species, with 400 genera and more than 5600 extant species [5,6,7]. Currently, eight subfamilies are recognized in this family [6]: Allantinae, Athaliinae, Beldoneinae, Blennocampinae, Heterarthrinae, Nematinae, Selandriinae, and Tenthredininae. However, the subfamilial classification of Tenthredinidae has indeed been a longstanding problem, and it is known that at least some of the traditionally or recently recognized subfamilies or tribes are most likely artificial or incorrectly placed units in both morphological and molecular analyses [6,7,8,9,10,11,12,13,14,15].

*Cladiucha* Konow, 1902, and *Megabeleses* Takeuchi, 1952, are two small tenthredinid genera that feed on plants of the family Magnoliaceae (angiosperms), and most species of these genera have great economic importance, mainly in Southeast Asia [16,17,18,19,20,21,22,23]. *Cladiucha* is more similar in some respects to Diprionidae and some Pergidae, harboring multiple antennomeres and serrate-type antennae in the female and biramous-type antennae in the male rather than the filiform antennae usually observed in other tenthredinids [24]. This genus was first placed in the subfamily Allantinae, considering the similarity in wing venation and other structural characteristics, but generated a new tribe, Cladiuchini, for this genus [9]. Takeuchi [18] and Abe and Smith [8] also placed these genera into the subfamily Allantinae. Wei [25] also recognized the tribe Cladiuchini, but placed *Cladiucha* and *Megabeleses* in a new subfamily, Megabelesesinae Wei, 1997, along with two additional new genera, *Tripidobeleses* Wei, 1997, and *Conobeleses* Wei, 1997. This representation was followed in the new system provided by Wei and Nie [26]. However, Taeger et al. [7] placed these four genera into the subfamily Allantinae. This systematic inconsistency indicates the requirement for more detailed studies to better understand the systematic position and evolutionary history of the subfamily Megabelesesinae using comprehensive molecular data from the current approaches.

Here, four complete mitogenomes were sequenced and characterized from the family Tenthredinidae: *Cladiucha punctata* Wei, 2020, *C. magnoliae* G.R. Xiao, 1994 [not 1993], *Megabeleses magnoliae* Wei, 2010, *M. liriodendrovorax* G.R. Xiao, 1993. These mitogenomes were compared with the previously reported tenthredinid mitogenomes for a better understanding of the characteristics of the mitogenome of Tenthredinidae. We also constructed a mitogenome dataset representing eight subfamilies of the family Tenthredinidae and five families of the superfamily Tenthredinoidea by adding 29 previously and 16 newly sequenced species (Appendix A). The dataset was generated for phylogenetic reconstruction of the family, applying several commonly used phylogenetic inference methods to overcome the systematic bias resulting from the non-stationarity of sequence evolution. A dated phylogeny was also rebuilt to assess the relationship between branching events and host shifts among the subfamilies.

## 2. Materials and Methods

### 2.1. Samples Analyzed

Specimens of four species were provided by Jiangxi Normal University, China, and specimen data were presented in Appendix A. Total genomic DNA was isolated from the hind leg of each ethanol-preserved specimen by the DNeasy Tissue Kit (Qiagen, Hilden, Germany) according to instructions provided by the manufacturer, and the isolated genomic DNA samples were quantified using a NanoDrop (Maestrogen, Inc., Waltham, MA, USA).

### 2.2. Mitogenome Sequencing, Annotation, and Analyses

The total genomic DNA extracts of four species were pooled and sequenced by the Illumina HiSeq 4000 next-generation sequencing (NGS) platform using 150-bp paired-end reads, conducted at Shanghai Majorbio Bio-pharm Technology Co., Ltd., China. Raw NGS reads of each species were firstly considered for quality control by FastQC v0.11.9 (http://www.bioinformatics.babraham.ac.uk/projects/fastqc) (accessed on 10 January 2021). The FASTQ files were imported into Geneious R11 (Biomatters Ltd, Auckland, New Zealand) [27], and the reads were then trimmed with BBDuk as implemented in Geneious R11. Duplicate reads and reads < 100 bp were filtered out to eliminate low-quality scores. The high-quality reads were then used to construct the mitogenome sequences of the four species. De novo assemblies of sequences were performed using MIRA assembler implemented in Geneious R11, and these assemblies were then mapped using the reference mitogenomes with ‘medium-low sensitivity’ parameters for each species. The assemblies constructed by these approaches were aligned, manually compared, and finally compiled into a single contig for each species. tRNA genes were found by their presumed secondary structure and anticodon sequence using the tRNAscan-SE server [28] and DOGMA [29] with the mito/chloroplast genetic code under the default search options. The boundaries and locations of the protein-coding genes (PCGs) and rRNA genes were identified by comparing the reported tenthredinid homologous gene sequences by ORF Finder (http://www.ncbi.nlm.nih.gov/gorf/gorf.html) (accessed on 10 January 2021) and by homology-based BLAST searches in GenBank. The precise ends of rRNA genes were predicted from the boundaries of the neighboring tRNA genes. The rRNA secondary structures were inferred by comparison with those of other reported sawfly species [30,31]. These predicted structures were visualized by VARNA v3-93 [32] and RNAviz 2.0.3 [33] with reference to the results of the CRW site [34]. Intergenic spacers and overlapping regions were inferred manually. The sequences of these four mitogenomes were deposited in GenBank under accession numbers MT295305-MT295306 and MW255939-MW255940. Summary statistics on the base composition, nucleotide substitution, and codon usage were analyzed by MEGA v6.0 [35]. Strand asymmetries were calculated using the following formulae: AT skew = (A − T)/(A + T) and GC skew = (G − C)/(G + C) [36].

### 2.3. Phylogenetic Analyses

Alignment and model selection. Phylogenetic analyses were carried out using a mitogenome dataset for the superfamily Tenthredinoidea: Thirty species (16 previously and 14 newly sequenced) from Tenthredinidae, representing eight known subfamilies, and 15 species (13 previously and two newly sequenced) from five other families, representing Argidae (three species), Cimbicidae (nine species), Diprionidae (one species), Heptamelidae (one species), and Pergidae (one species). One species from the superfamily Xyeloidea was included as an outgroup (Appendix A). Mitogenomes of 29 previously reported tenthredinoid species and one outgroup were retrieved from the NCBI annotated database using Geneious R11. Each PCG was aligned under codon-based multiple alignments with ClustalW [37] as implemented in MEGA v6.0. The RNA genes were aligned as DNA using the MAFFT algorithm [38] as implemented in Geneious R9. Each obtained alignment was then concatenated using SequenceMatrix v.1.7.8 [39]. The optimal partition strategy and best fit evolutionary model of each partition were designated by PartitionFinder v1.1.1 [40], applying the Bayesian information criterion (BIC) and the ‘greedy’ algorithm based on branch lengths estimated as ‘unlinked’. The data blocks were defined by genes and codons to generate an input configuration file with 63 (with all codon positions) and 50 (without 3rd codon positions) partitions. The best partitioning schemes and related models were used in all subsequent phylogenetic analyses (Appendix A).

Test of substitution saturation. The genetic saturation levels of different codon positions and genes were calculated by correlation coefficient analysis implemented in R core packages [41], comparing the distances estimated by applying the best fit evolutionary model GTR + G + I proposed by jModelTest2 with the uncorrected p-distances [42]. The distance values were measured with PAUP v4.0b10 [43].

Phylogenetic inference. The phylogenetic reconstructions were performed with four different datasets: (i) The 13 PCGs with all codon positions plus 22 tRNAs and two rRNAs (P123RNA) (16,719 bp); (ii) the 13 PCGs with the 1st and 2nd codon positions plus 22 tRNAs and two rRNAs (P12RNA) (12,811 bp); (iii) the PCGs with all codon positions excluding three genes plus 22 tRNAs and two rRNAs (P123RNAexc3genes) (15,871 bp), and (iv) the PCGs with the 1st and 2nd codon positions excluding three genes plus 22 tRNAs and two rRNAs (P12RNAexc3genes) (12,150 bp). The fourth dataset was generated by considering the result of the substitution saturation test, which indicated lower degrees of correlation between all codon positions of PCGs and the first and second codon positions of *atp8*, *nad4l*, and *nad6* (Appendix A). Maximum likelihood (ML) and Bayesian inference (BI) approaches were used for each dataset to infer whether the datasets were sensitive to phylogenetic inference methods. ML analyses were conducted in RAXML v8.0.9 [44] as implemented in Geneious R11, using the suggested GTR + G + I substitution model by PartitionFinder v1.1.1 for each nucleotide partition under 1000 bootstrap replicates. BI analyses were performed for unlinked branch lengths of each partition scheme under MrBayes v3.2.2 [45]. Distribution of posterior parameters were estimated in two independent runs with four Markov chains (three cold, one heated) based on 10 million generations and sampling every 1000 generations. The log-likelihood files produced by each run were assessed considering ESS values equal to or greater than 200 for all priors using Tracer v1.7 [46]. After the assessment, the first 25% of trees sampled in each run were scrapped as burn-in, and a majority-rule consensus tree (BI tree) was constructed from the remaining trees.

Divergence time estimation. The divergence times among the tenthredinids were estimated in BEAST v1.8.3 utilizing the dataset of P123RNA [47]. The uncorrelated relaxed lognormal clock was preferred because the dataset was found to be “not clock-like” (likelihood ratio test: −ln + c 210,126.327, −ln−c 209,813.739, d.f. = 15, *p* = 5.14E–92). Yule speciation tree priors were applied to model rate variation among species [48]. To estimate divergence times, a second calibration point was used: The age of (i) the split between the clade of Argidae-Pergidae and all remaining Tenthredinoidea (a mean of 175 Ma and ranging from 150–200 Ma) and (ii) the Argidae and Pergidae split (a mean of 135 Ma and ranging from 110–160 Ma) based on previous studies [4,49,50]. The run was performed using 100 million generations with samples taken every 10,000 generations, and the result was evaluated to assess convergence and confirm effective sample sizes (ESS > 200) in Tracer v1.6. The maximum clade credibility of the trees was quantified using TreeAnnotator v1.8.3 [47] after discarding 25% of the samples as burn-in. The obtained tree was then visualized in FigTree v1.4.2 [51].

## 3. Results

### 3.1. Mitogenome Architectures and Nucleotide Compositions

The complete mitogenomes of four species representing two genera were sequenced and characterized: *C. magnoliae* (15,761 bp), *C. punctata* (16,187 bp), *M. magnoliae* (16,219 bp), and *M. liriodendrovorax* (15,466 bp) (Figure 1, Appendix A). Each mitogenome consisted of a typical set of 37 genes: Thirteen PCGs, twenty-two tRNAs, two rRNAs, and an A + T rich region. The observed length variation among these mitogenomes was primarily due to variation in the A + T-rich region (Appendix A). The genes were mostly located on the majority J strand, except for four PCGs (*nad1*, *nad4*, *nad4l*, and *nad5*), two rRNAs, and eight tRNAs. The mitochondrial gene orders of the four species were entirely conserved (Figure 1). However, their arrangements were slightly different from those of the inferred ancestral pancrustacean mitogenome (*Daphnia pulex*), with the occurrence of several tRNA gene rearrangements. The first rearrangement event was found in the IQM gene cluster; here, the *trnQ* gene was inverted, and the *trnM* and *trnI* genes were transposed, moving upstream and downstream of the *trnQ* gene, respectively (arranged as MQI, Figure 1). The second event was a reverse transposition of the *trnR* gene upstream of the *trnF* gene in the ARNS1EF gene cluster (arranged as ANS1ERF, Figure 1).

The nucleotide compositions of these mitogenomes were biased toward A and T, ranging from 80.7% (*M. liriodendrovorax*) to 83.1% (*C. punctata* and *C. magnolia*) (Appendix A). Excluding the A + T-rich regions, the highest AT content was found in rRNA genes of *C. punctata* (85.5%), and the lowest was observed in the PCGs of *M. liriodendrovorax* (79.5%). The skew metrics in the whole mitogenomes were different with respect to the AT skew, but similar in terms of the GC skew. A more pronounced A bias was observed in *Megabeleses*, with an average AT skew of 0.033, than in *Cladiucha* (0.013). However, a deviation was found in the PCGs: T-skewed in all species (with an average AT skew of −0.113) and G-skewed in *C. magnoliae* (0.006), *M. liriodendrovorax* (0.005), and *M. magnoliae* (0.026). This deviation probably results from T bias in the 2nd codon position and G bias in the 1st codon position. All PCGs and rRNA genes were similar in length (Appendix A). However, a variation was observed in cox2, with six codon differences between *M. magnoliae* (225 aa) and the other species (227 aa). The estimated initiation codons commonly encoded either isoleucine or methionine (ATN), and termination codons were conserved, except for *cob* in two *Megabeleses* species and *nad4* in two *Cladiucha* species and *M. magnoliae*, which all use partial termination codons (T-). A significant correlation was observed between nucleotide composition and codon preference (Figure 2, Appendix A). UUA-Leu had the highest relative synonymous codon usage (RSCU), with an average value of 3.53, and all the remaining codons with RSCU greater than 2.00 had T or, particularly, A in the third codon position (Figure 2).

All the predicted tRNAs had standard anticodons, ranging in size from 60 bp (*trnV* in *M. liriodendrovorax*) to 71 bp (*trnK* in *Megabeleses*, *trnT* in *C. magnoliae*) and folding into canonical clover-leaf structures, except for all *trnS1* in all four species and *trnV* in two *Megabeleses* species (Figure 3). The identical location of the rRNA genes was found in all species, with *rrnS* between the *trnV* and A + T-rich regions and *rrnL* positioned between *trnL1* and *trnV* (Figure 1, Appendix A). The putative secondary structures of rrnL and rrnS genes in four species were consistent with the models suggested for other insects, consisting of 49 helices from five structural domains (domain III is absent as in other arthropods) (Figure 4 and Figure 5). Variation was observed in the total length of intergenic regions among the species. The total length of the intergenic regions was 123 bp (at 19 different locations) in *M. liriodendrovorax*, 253 bp (at 17 locations) in *C. magnoliae*, 324 bp (at 17 locations) in *C. punctata*, and 461 bp (at 19 locations) in *M. magnoliae* (Appendix A). Homology searches for these intergenic regions showed no significant similarity with any identified nucleotide sequence. The length and location of overlaps between adjacent genes were similar in both *Cladiucha* (between 1 and 7 bp) and *Megabeleses* (between 1 and 8 bp) (Appendix A).

### 3.2. Phylogeny of Tenthredinidae

Eight phylogenetic reconstructions (four datasets by two inference approaches) recovered four different tree topologies (Appendix A and Table 1). The obtained topologies were sensitive to both inference approach and dataset differences, and the support values were mostly lower for ML trees than for the BI of the corresponding dataset (Figure 6 and Appendix A). The exclusion of third codon positions did not affect the topology of the recovered phylogenetic tree, but the tree was sensitive to the saturated PCGs (*atp8*, *nad4l*, and *nad6*; Appendix A). The most frequently obtained tree was recovered from the datasets of P123RNA and P12RNA, with both ML and BI approaches finding the same topology. This tree topology was also obtained from the dataset of P123RNAexc3genes applying just the ML inference method. The recovered tree supported the monophyly of Pergidae, Argidae, Diprionidae, and Cimbicidae, while a paraphyletic relationship was observed in Tenthredinidae, with the placement of Athaliinae (Figure 6). Argidae and Pergidae formed a monophylum that was sister to the remaining taxa. Next, *Athalia* (currently classified in the tenthredinid subfamily Allantinae) was separated from a large clade consisting of the representatives of Diprionidae, Cimbicidae, Heptamelidae, and Tenthredinidae with strong nodal support [posterior probability (PP) = 1.00 and ML = 100%] (Figure 6). A sister-group relationship was supported between Diprionidae and Cimbicidae (PP = 1.00, ML = 73%), and this clade was also identified as sister to Heptamelidae + Tenthredinidae (PP = 1.00, ML = 78%). At the family level, the topology recovered P12RNAexc3genes under both inferences and P123RNAexc3genes under BI supported a relationship of (Heptamelidae + (Diprionidae + Cimbicidae)) + Tenthredinidae (PP = 1.00, ML = 61%) (Appendix A, Table 1). Within Tenthredinidae *s. str.* (except for a rogue taxon *Athalia*), all analyses supported the monophyly of the subfamilies Nematinae, Selandriinae, and Tenthredininae with high support, while Allantinae, Heterarthrinae, and Blennocampinae were found to be paraphyletic (Figure 6 and Appendix A). Nematinae was robustly placed as the most basal tenthredinid subfamily with strong nodal support (PP = 1.00, ML = 100%), except for the topology recovered under P12RNAexc3genes under BI, which was obtained a sister-group relationship between Nematinae and Selandriinae. After the divergence of Nematinae, Selandriinae was recovered as a sister to all remaining tenthredinids (Figure 6 and Appendix A). Next, the four newly sequenced species representing *Cladiucha* and *Megabeleses* (currently in Allantinae) formed a distinct lineage with high support values (PP = 1.00, ML = 98%). They were sisters to the representatives of Blennocampinae, Heterarthrinae, Tenthredininae, and the rest of Allantinae (Figure 6 and Appendix A). For the remaining four subfamilies, a sister-group relationship between Heterarthrinae, Blennocampinae, and *Hemibeleses tianmunicus* (Allantinae) forming a paraphyletic grade, and Tenthredininae + Allantinae *s. str* with strong nodal supports (PP = 1.00, ML = 78%), except for the topology recovered under P12RNAexc3genes under ML, which Tenthredininae had a basal placement (ML = 88%) (Figure 6 and Appendix A, Table 1).

### 3.3. Divergence Time Estimations at the Subfamily Level

The inferred divergence and diversification time estimates of tenthredinids were based on the dataset of P123RNA and a secondary-calibrated relaxed molecular clock (Figure 7). According to the dated tree, the stem-group age of Tenthredinidae was estimated to be 114.46 Ma (94.10–134.71) in the Early Cretaceous, which also corresponds to the split of Heptamelidae and Tenthredinidae. Diversification of Tenthredinidae occurred between the end of the Early Cretaceous and the late Miocene (Figure 7). The oldest division in the tenthredinids is the subfamily Nematinae, with an estimated age of 104.39 (85.63–122.66) Ma. The stem age of the *Cladiucha* and *Megabeleses* lineages was dated at 95.13 (83.64–118.87) Ma in the Early–Late Cretaceous boundary, while the split between these genera was estimated to have occurred at the beginning of the Eocene at 53.34 (39.92–67.67) Ma (Figure 7). Within this lineage, the divergence time of *M. magnoliae* and *M. liriodendrovorax* corresponded to the Oligocene [31.76 Ma, (21.57–42.62 Ma)], while the split of *C. magnoliae* and *C. punctata* was the most recent and was dated to 10.98 (6.97–15.49) Ma, corresponding to the late Miocene.

## 4. Discussion

### 4.1. Mitogenome Organization of Cladiucha and Megabeleses

Four complete and twelve nearly complete mitogenomes of Tenthredinidae have been reported so far [14,53,54,55,56,57,58,59,60,61]. Here, the complete mitogenomes of four species representing *Cladiucha* and *Megabeleses* were sequenced for the first time, and their mitogenome characteristics were summarized at the subfamily level by comparing them with the published and/or unpublished mitogenomes representing the family. All sequenced genomes are highly conserved in size, nucleotide content, codon usage, and secondary structures of tRNAs and rRNAs (Figure 2, Figure 3, Figure 4 and Figure 5, Appendix A). Their mitogenome sizes fit well within the range reported for tenthredinid mitogenomes, as well as for Symphytan mitogenomes, ranging from 15,108 bp in *Hemathlophorus brevigenatus* (MW632125 in GenBank) to 20,370 bp in *Trachelus iudaicus* [62]. Similar to the reported rearrangement hotspot points in hymenopteran mitogenomes [63,64], the most rearrangements were frequently observed in the IQM, ARNS1EF, and WCY gene clusters (Figure 1), possibly related to illicit priming of mitochondrial replication and/or illegitimate intramitochondrial recombination [65,66]. tRNA rearrangements have been most frequently reported; the mitogenomes of the derived hymenopteran suborder Apocrita appear to be more prone to rearrangements than Symphytan mitogenomes, but the observed pattern in tenthredinid mitogenomes, as well as other reported Symphytan mitogenomes, supports the frequent rearrangement of tRNA genes in Symphyta [31,60,62,67,68,69]. Here, the same type of architecture of tRNA rearrangement in the IQM (arranged as MQI) and ARNS1EF (arranged as ANS1ERF) gene clusters in the sequenced mitogenomes of both *Cladiucha* and *Megabeleses* species may provide further evidence of molecular synapomorphy for this lineage (Figure 1). These derived gene rearrangements were also not found in other tenthredinid taxa, indicating that they occurred during the divergence of this lineage, and therefore, supported their recognition as a subfamily [25].

### 4.2. Phylogenetic Placement of Cladiucha and Megabeleses

Among the known Symphytan families, the internal phylogeny of Tenthredinidae has attracted much attention because of the unstable phylogenetic placement of the subfamilies, as well as their monophyly within the family depending on the combination or absence of enough morphological characters, relatively narrow range of taxonomic coverage and a limited number of easily accessible sequence datasets [10,12]. To elucidate relationships within the family, previous phylogenetic studies based on morphological characters and/or recent comprehensive molecular analyses have provided several conclusions. (i) The genus *Athalia* was moved to the subfamily Athaliinae rather than Allantinae [10], but its taxonomic status remains unstable. This conclusion was mainly based on the placement of *Athalia*, which was placed as either sister to all other members of the family [10,70,71] or outside of the tenthredinids, with changing placement among the families Cimbicidae, Diprionidae, Heptamelidae, and Tenthredinidae [49,72,73,74,75]. (ii) The genus *Heptamelus* or the tribe Heptamelini should be excluded from Selandriinae [12]. Moreover, the genera *Heptamelus*, *Carinoscutum*, *Parahemitaxonus*, and *Pseudoheptamelus* have recently been recognized as members of a small new family, Heptamelidae [6,10]. (iii) The monophyly of Nematinae and Tenthredininae was supported in most studies, but the obtained topologies can also introduce uncertainty in the placement of the remaining traditional subfamilies [10,12,49].

Here, the internal phylogeny of the family was produced with a mitogenome dataset, including 13 PCGs (three saturated genes were excluded in some analyses, Appendix A), two rRNA genes, and 22 tRNA genes (15,365 bp in length), but with a relatively low number of species (30 species representing eight subfamilies). Despite the fact that the present phylogeny suggests a complex evolutionary history, the recovered branching pattern with high nodal support was mostly compatible with previously reported phylogenies (Figure 6) [10,12,49,72,73,75]. The phylogenetic placement of the genus *Athalia*, outside of Diprionidae, Cimbicidae, Heptamelidae, and Tenthredinidae, obviously did not support its current classification under the subfamily Athaliinae and required reassessment to provide a solid foundation of its taxonomic position. The sister-group relationship between Heptamelidae and Tenthredinidae was supported in most of the analyses (Figure 6), as described by Boevé et al. [70]. At the same time, the placement of Heptamelidae as a sister taxon to a clade comprising Diprionidae, Cimbicidae, and Tenthredinidae was also recovered by Malm and Nyman [10] (Table 1, Appendix A). Within Tenthredinidae, the present phylogeny is broadly congruent with previously reported relationships, supporting the monophyly of the subfamilies Nematinae, Selandriininae, and Tenthredininae (Figure 6) [10,11,49]. However, the formation of the paraphyletic grade comprising the representatives of Heterearthrinae, Blennocampinae, and *Hemibeleses* (Allatinae) indicates that on the one hand, these subfamilies need to revision in their current classifications, but on the other hand, this suggestion would be premature to recommend a new subfamilial classification for Tenthredinidae until a more comprehensive phylogeny becomes available, including more genera and/or species representatives of these subfamilies. But even so, *Cladiucha* and *Megabeleses* were consistently identified as monophyletic and outside of Allantinae (Figure 6 and Table 1). The synapomorphy of Megabelesesinae is as follows: The female lance has a long and broad membranous lobe and approaches the upper 0.3–0.5 of the lancet (Figure 7). Another possible synapomorphic character is the vein 1m-cu being very close to the base of vein Rs. Some shared morphological characters among the species of these genera, but not in the species of Allantinae, also support Megabelesesinae as a plesiomorphic: (i) The mandibles are simple and bidentate, (ii) the head is not distinctly enlarged behind the eyes, and (iii) the anterior lobe of the pronotum is very narrow. Their phylogenetic placement as a distinct lineage, in conjunction with (i) the morphological synapomorphy discussed above; (ii) the shared molecular features (occurrence of the same type of tRNA rearrangement, arranged as MQI and ANS1ERF in their sequenced mitogenomes; Figure 1, Appendix A); and (iii) host specialists feeding on Magnoliaceae, may provide evidence supporting the proposal of the subfamily Megabelesesinae.

### 4.3. Divergences of Tenthredinid Subfamilies through Time

The stem group age and crown age of Tenthredinidae are estimated to be older than those reported by Peters et al. [50], but broadly congruent with previously reported dated trees [4,49,73]. A close association between the host preferences and divergence times of the subfamilies was observed in the dated tree (Figure 7), supporting the suggestion that the diversification rate increments in tenthredinids can be directly related to dominant colonization of flowering plants (angiosperms) following their initial appearance by the Early Cretaceous between approximately 130 and 100 Ma [4,76,77]. The estimated crown ages of the subfamilies also suggest that the major divergence within the family corresponds to the Cenomanian/Turonian (estimated at approximately 95 Ma) and Cretaceous–Paleogene boundaries (estimated at approximately 70 Ma). These important geological intervals, recognized as severe biotic crises in the history of life [78], played an unignorable role in influencing the floral compositions, with minor changes in the diversification of nonflowering seed plants (gymnosperms), significant extinction events in spore-bearing plants (pteridophytes) and rapid increases in the origin and diversification rates of angiosperms [77]. The sudden and rapid changes in the global biodiversity patterns during these periods most likely triggered diversification and colonization of new host plants for tenthredinid sawflies [4].

The present dated tree also suggests that the divergence and diversification times of *Cladiucha* and *Megabeleses* nearly coincide with the divergence times of their host plants (Figure 7). The members of these genera constituting a distinct lineage, Megabelesesinae, appear to be more special with a relatively narrow host range, feeding on the members of the family Magnoliaceae [20,24,79]. Based on fossil records and evidence from molecular phylogenetic studies, this family has generally been considered one of the relatively primitive families of flowering plants with woody members [80,81]. The origin of Magnoliaceae and split time between two subfamilies—namely, Magnolioideae and Liriodendroideae, are estimated to correspond to the Early–Late Cretaceous boundary, occurring approximately 93.5 Ma [80], 100 Ma [82], or 113 Ma [83]. However, the crown ages of Magnolioideae and Liriodendroideae are dated to approximately 42 Ma or 54 Ma and 28 Ma or 15 Ma, respectively [80,82]. The subfamily and species divergence of the family Magnoliaceae mostly matches the divergence of *Cladiucha* and *Megabeleses*, as well as their species diversification (Figure 7). However, concomitant radiation does not necessarily imply a reciprocally causative evolutionary scenario—at best, our data do not exclude the hypothesis that speciation of herbivorous insects was triggered by host shifts depending on the radiation of their host plants.

## Figures and Tables

**Figure 1 insects-12-00495-f001:**
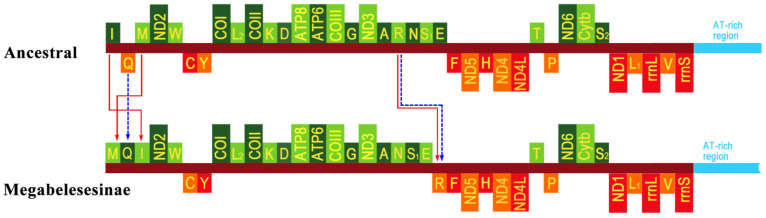
Mitochondrial genome organization of Megabelesesinae with reference to the ancestral type of insect mitochondrial genomes. Genes transcribed from the J- and N-strands are shown in green and orange, respectively. The A + T-rich region is indicated by blue, and tRNA genes are labeled by their single-letter amino acid code.

**Figure 2 insects-12-00495-f002:**
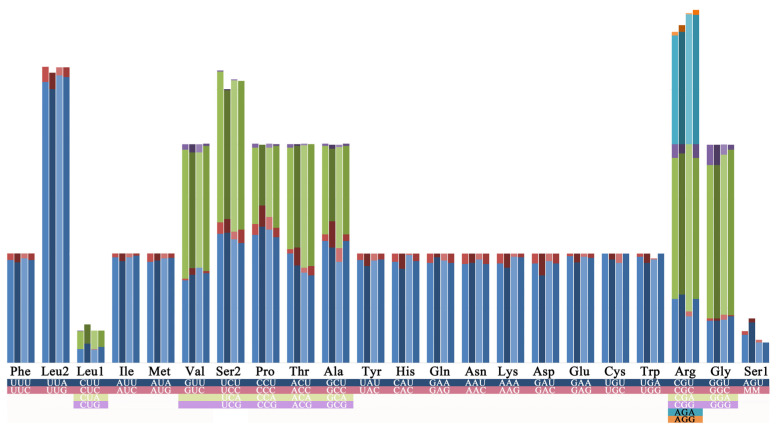
Relative synonymous codon usage (RSCU) of the Megabelesesinae mitogenomes. The RSCU values are shown as the accumulative bar diagrams; from left to right, column shows *Megabeleses magnoliae*, *M. liriodendrovorax*, *Cladiucha punctata*, and *C. magnoliae*, respecitively. The accumulative bar diagrams of codon families are provided on the *x*-axis.

**Figure 3 insects-12-00495-f003:**
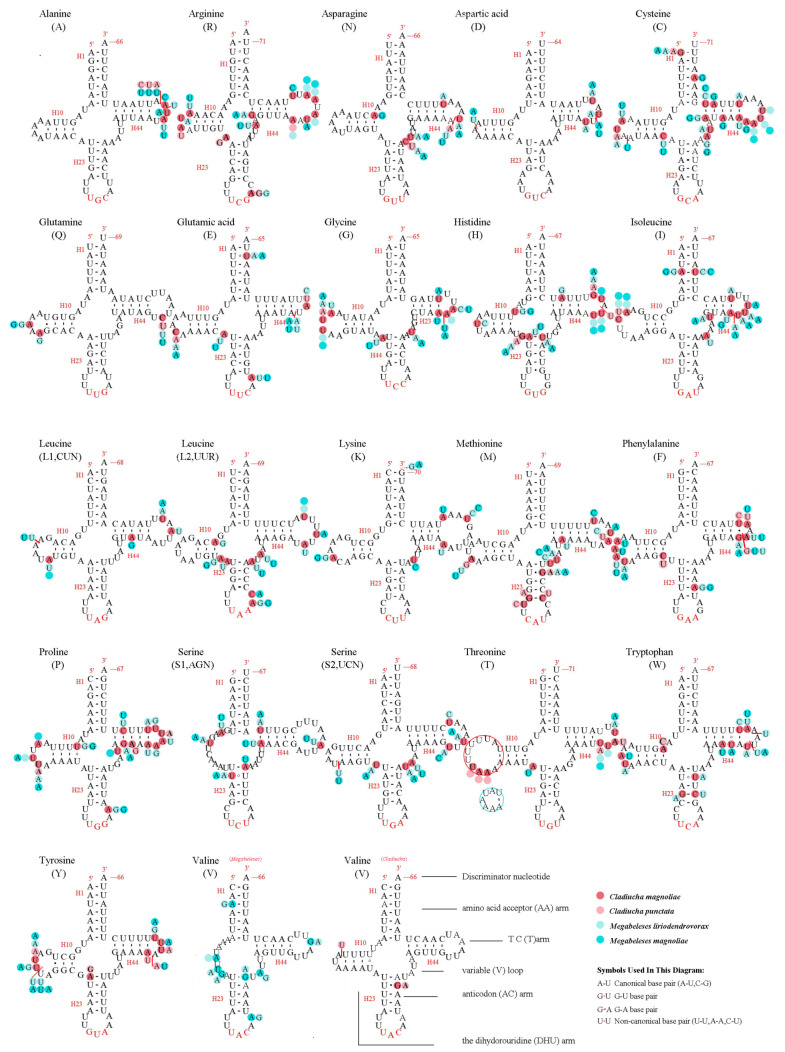
Predicted secondary structures of 22 tRNA genes of *Cladiucha*. Dashes indicate Watson-Crick base pairs, and dots indicate G-U base pairing.

**Figure 4 insects-12-00495-f004:**
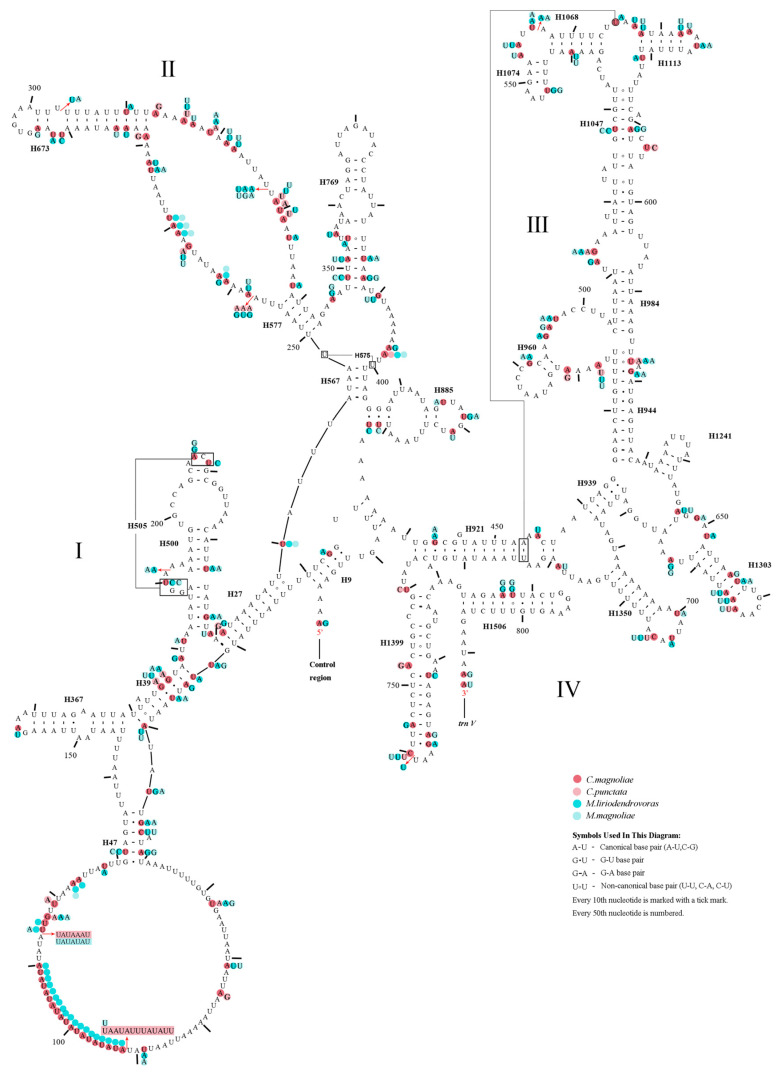
Predicted rrnS secondary structure in the *Cladiucha* mitochondrial genome. The numbering of helices follows Gillespie et al. [52]. Roman numerals refer to domain names. Tertiary inter-actions and base triples are connected by continuous lines. *C. magnoliae* as a basemap and base change among *Cladiucha* species are presented in circles with red (*C. magnoliae*) and pink (*C. punctata*) colors.

**Figure 5 insects-12-00495-f005:**
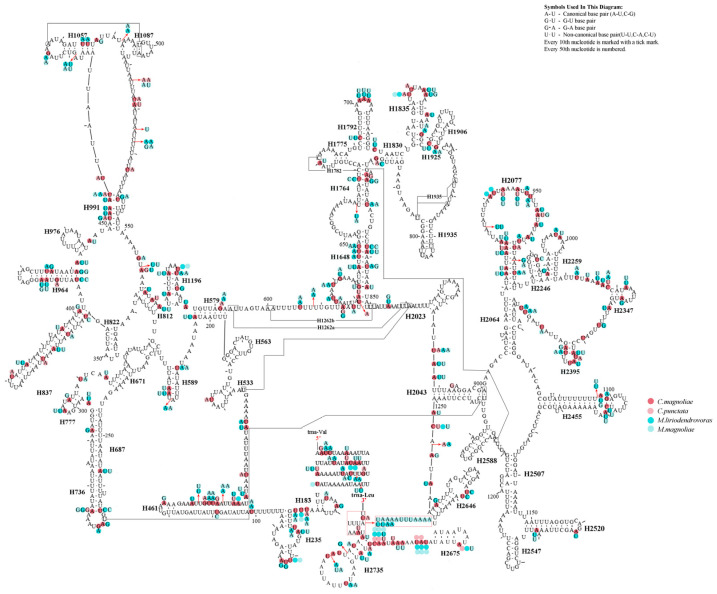
Predicted rrnL secondary structure in the *Cladiucha* mitochondrial genome. The numbering of helices follows Gillespie et al. [52]. Roman numerals refer to domain names. Tertiary inter-actions and base triples are connected by continuous lines. *C. magnoliae* as a basemap and base change among *Cladiucha* species are presented in circles with red (*C. magnoliae*) and pink (*C. punctata*) colors.

**Figure 6 insects-12-00495-f006:**
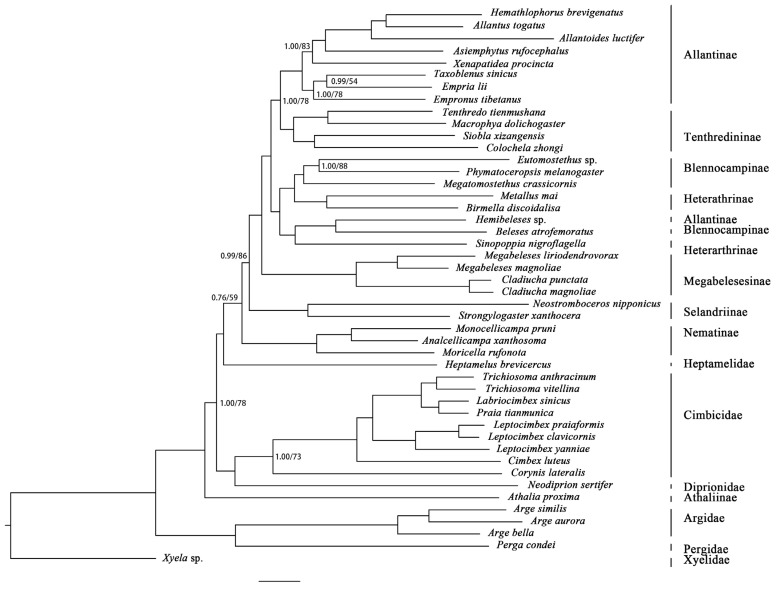
Phylogenetic tree recovered using the dataset P123RNA using Bayesian inference and maximum likelihood methods. Only support values <1.0 (posterior probabilities) and <100% (bootstraps) are shown.

**Figure 7 insects-12-00495-f007:**
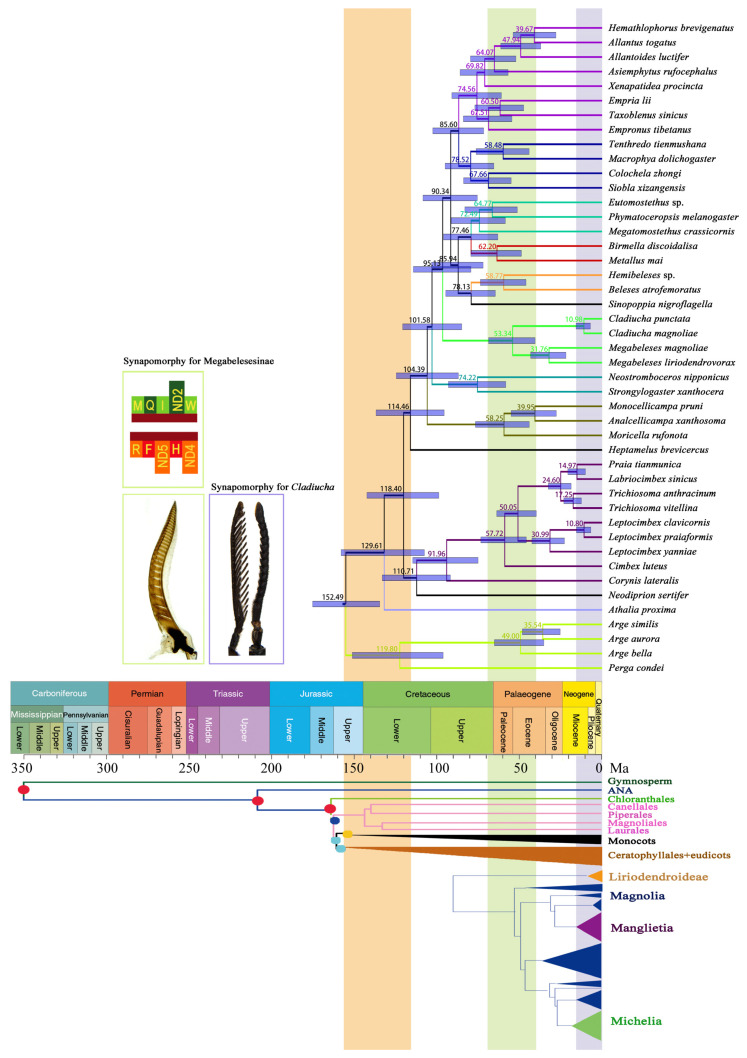
Dated phylogeny constructed with P123RNA dataset in BEAST v1.8.3. The axis on the bottom refers to million years and shows the geological time. The blue bars on the nodes represent 95% of the high posterior density of divergence times obtained from the MCMC tree. The divergence times of each node obtained from the MCMC tree analysis are written in black on the nodes. The yellow shaded area delineates the arising of Tenthredinoidea, while the green and purple shaded areas delineate the arising of Megabelesesinae and *Cladiucha*, respectively, depending on the minimum or maximum ages. The traits in green and purple boxes are their synapomorphies, respectively. The phylogenetic trees given at the bottom are the angiosperm tree and the Magnolia tree, respectively.

**Table 1 insects-12-00495-t001:** Summary of the phylogenetic relationships recovered by different datasets and inference approaches.

**Datasets**		**Within Tenthredinoidea**		**Within Tenthredinidae**	
	**Inference Methods**	**BI**	**ML**	**BI**	**ML**
	P123RNA	(P + A) + (Ath + ((D + C) + (H + T)))	(P + A) + (Ath + (D + C) + (H + T)))	N + (S + (M + ((Ht + (Al + B)) + (Ht + B)) + (Th + Al))))	N + (S + (M + ((Ht + (Al + B)) + (Ht + B)) + (Th + Al))))
	P12RNA	(P + A) + (Ath + ((D + C) + (H + T)))	(P + A) + (Ath + ((D + C) + (H + T)))	N + (S + (M + ((Ht + (Al + B)) + (Ht + B)) + (Th + Al))))	N + (S + (M + ((Ht + (Al + B)) + (Ht + B)) + (Th + Al))))
	P123RNAexc3genes	(P + A) + (Ath + ((H + (D + C)) + T))	(P + A) + (Ath + ((D + C) + (H + T)))	N + (S + (M + ((Ht + (Al + B)) + (Ht + B)) + (Th + Al))))	N + (S + (M + ((Ht + (Al + B)) + (Ht + B)) + (Th + Al))))
	P12RNAexc3genes	(P + A) + (Ath + ((H + (D + C)) + T))	(P + A) + (Ath + ((H + (D + C)) + T))	N + (S + (M + (((Ht + (Al + B)) + (Ht + B)) + (Th + Al)))	N + (S + (M + (Th + (((Ht + (Al + B)) + (Ht + B)) + Al))))

ML, maximum likelihood; BI, Bayesian inference; P123, all codon positions of PCGs; P12, 1st and 2nd codon positions of PCGs; RNA, nucleotide sequences of rrnS, rrnL and tRNA genes; ‘exc3genes’ indicates that *nad4l*, *nad6* and *atp8* excluded from analyses; P, Pergidae; A, Argidae; Ath, Athaliinae; D, Diprionidae; C, Cimbicidae; H, Heptamelidae; T, Tenthredinidae; N, Nematinae; S, Selandriinae; M, Megabelesinae; Ht, Heterarthrinae; Al, Allantinae; B, Blennocampinae; Th, Tenthredininae. Please see Appendix A for details.

## Data Availability

The data presented in this study are openly available in Science Data Bank at DOI: 10.11922/sciencedb.00702.

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
