# Peer review of "Mitochondrial Phylogenomics of Tenthredinidae (Hymenoptera: Tenthredinoidea) Supports the Monophyly of Megabelesesinae as a Subfamily"

_insects, 2021, doi:10.3390/insects12060495_

Round 1

Reviewer 1 Report

Dear Dr. Wei,

I have carefully read your manuscript entitled "Mitochondrial phylogenomics of Tenthredinidae (Hymenoptera: Tenthredinoidea) supports the monophyly of Megabelesesinae as a new subfamily". I believe it contains new important information on mitogenomes of four members of the family which belong to two related genera. In turn, these genera form a newly proposed subfamily. The paper is generally well-written. However, I suggest some technical corrections to the manuscript (please see the attached file). Please also correct the name "Strongylogaster xanthocera" in the Supplementary materials.

Yours sincerely,

Author Response

Comments to the Author

I have carefully read your manuscript entitled "Mitochondrial phylogenomics of Tenthredinidae (Hymenoptera: Tenthredinoidea) supports the monophyly of Megabelesesinae as a new subfamily". I believe it contains new important information on mitogenomes of four members of the family which belong to two related genera. In turn, these genera form a newly proposed subfamily. The paper is generally well-written. However, I suggest some technical corrections to the manuscript (please see the attached file). Please also correct the name "Strongylogaster xanthocera" in the Supplementary materials.

Replies to general comments

As accordance with reviewer’s technical and linguistic corrections on the marked-up copy of the manuscript, we have revised the manuscript thoroughly. The name of "Strogylogaster xanthocera" was also revised as “Strongylogaster xanthocera” in all phylogenetic trees in the Supplementary materials.

Reviewer 2 Report

I recommend publishing the paper, but the authors should be more careful making strong statements. The subfamily Megabelesesinae does not seem justified because there are too few tenthredinid genera included. The authors can certainly discuss about validity of this subfamily, but should also discuss if this could turn out to be wrong. There are so many other genera described that could be relevant for subfamilial classification of Tenthredinidae (Dinax, Empronus, Eriocampa, Beleses, Hemibeleses, Conaspidia etc). The statements about divergence times in general and of Megabelesesinae in relation to Magnoliaceae in particular are also questionable. The estimates of sawfly and angiosperm divergence times vary so much in the published literature (not without reason, because estimating these times is unreliable based on current knowledge) that you can make them fit with almost any evolutionary scenario you choose based on cherry picked studies. Again, the authors can discuss various hypotheses, but should acknowledge the limitations of divergence time estimates and discuss alternatives. Also, it should be mentioned that divergence times estimated from mitochondrial genomes tend to be generally older than times estimated from nuclear genomes, which is also relevant here. The authors estimated the times using mt genomes, but Nyman et al (2019) who have generally younger divergence times, had mostly nuclear genes.

One other thing. The documentation of analysed samples (specimen collection data, ID numbers etc) is missing (for some specimens there is some data in GenBank, but inconsistently). As much data should be given, at least provided in the publication (preferably also included in GenBank). Include a supplementary table perhaps?

Further comments and corrections in the attached pdf.

Round 2

Reviewer 2 Report

Line 31: replace “excellent” with “apparent”, as the authors themselves get somewhat different divergence times based on different datasets

Line 32: delete “strongly”

Line 106: replace “non-stationarity sequence evolution” with “non-stationarity of sequence evolution”.

Line 331: replace “Selandrinae” with Selandriinae.

Line 382-383: “The green and yellow shaded areas delineate the arising of Tenthredinoidea and Cladiucha, respectively”.

Shouldn’t it be “The yellow and green shaded areas delineate the arising of Tenthredinoidea and Cladiucha, respectively”?

Line 473: delete ”strongly”, because sampling for Nematinae and Selandriinae (correct also the name) is too small to claim this.

Line 486: replace “plesiomorphy” with “plesiomorphic”

The authors should mention that they themselves get different divergence times based on different datasets illustrating the challenge of getting reliable divergence times in general.

Author Response

Line 31: replace “excellent” with “apparent”, as the authors themselves get somewhat different divergence times based on different datasets

Reply: done as suggested

Line 32: delete “strongly”

Reply: removed as suggested

Line 106: replace “non-stationarity sequence evolution” with “non-stationarity of sequence evolution”.

Reply: replaced as suggested

Line 331: replace “Selandrinae” with Selandriinae.

Reply: revised as suggested

Line 382-383: “The green and yellow shaded areas delineate the arising of Tenthredinoidea and Cladiucha, respectively”.

Shouldn’t it be “The yellow and green shaded areas delineate the arising of Tenthredinoidea and Cladiucha, respectively”?

Reply: revised as “The yellow shaded area delineate the arising of Tenthredinoidea. The green and purple shaded areas delineate the arising of Megabelesesinae and Cladiucha, respectively, depending on the minimum or maximum ages. The traits in green and purple boxes are their synapomorphies, respectively.”. Please see the legend of Figure 7.

Line 473: delete ”strongly”, because sampling for Nematinae and Selandriinae (correct also the name) is too small to claim this.

Reply: removed as suggested

Line 486: replace “plesiomorphy” with “plesiomorphic”

Reply:  replaced as suggested

The authors should mention that they themselves get different divergence times based on different datasets illustrating the challenge of getting reliable divergence times in general.

Reply: The reviewer is right in his/her suggestion. But we cannot use different dataset to estimate the divergence times. Also we did not perform any comparison on the result of estimation analyses.

This manuscript is a resubmission of an earlier submission. The following is a list of the peer review reports and author responses from that submission.